# Novel Bathing Assist Device Decreases the Physical Burden on Caregivers and Difficulty of Bathing Activity in Care Recipients: A Pilot Study

**Kenji Kato [1],*,†, Keita Aimoto [2],†, Koki Kawamura [2], Tatsuya Yoshimi [1], Naoki Itoh [2] and Izumi Kondo [3]**

1   Laboratory of Clinical Evaluation with Robotics, Assistive Robot Center, National Center for Geriatrics and Gerontology, 7-430, Morioka, Obu 474-8511, Japan
2   Department of Rehabilitation Medicine, National Center for Geriatrics and Gerontology, 7-430, Morioka, Obu 474-8511, Japan
3   Assistive Robot Center, National Center for Geriatrics and Gerontology, 7-430, Morioka, Obu 474-8511, Japan
*   Correspondence: kk0724@ncgg.go.jp; Tel.: +81-562-462311
†   These authors contributed equally to this work.

**Featured Application: Reducing Physical Burden for Caregivers by Use of a Bathing Assist Device.**

**Abstract:** The purpose of this study was to investigate whether the use of a bathing assist device, "Bath Assist" (BA), could reduce the physical burden on caregivers providing bathing assistance and also alleviate the difficulty of bathing activities for care recipients. BA is a robotic device that is attached to the bathtub. The seat can be raised and lowered using the water pressure from the shower tap. The four caregivers and four care recipients were paired one-to-one to simulate bathing activity, and the physical burden on each caregiver during bathing assistance was evaluated with and without the use of BA by observation, questionnaire, and wireless surface electromyography. For caregivers, BA transformed the task of assisting care recipients into and out of the bathtub into a monitoring activity. Additionally, the muscle activity of the caregivers' lumbar region, trunk, and upper limbs, which are used to assist care recipients out of the bathtub, was significantly reduced when using BA. Questionnaires confirmed that the use of BA reduced the physical burden on caregivers. These results indicate that BA has the potential to reduce both the physical burden on caregivers during bathing assistance and the difficulties care recipients experience during bathing.

**Keywords:** bathing care; assistive technology; electromyography; physical burden; caregiver

## 1. Introduction

Lower back pain is one of the most common musculoskeletal disorders among nurses and other healthcare workers worldwide [1–5]. In various surveys of caregivers in Japan, 55–80% of caregivers complain of back pain, and back pain is recognized as a serious occupational health and safety problem for caregivers [6–8]. The occurrence of back pain as well as excessive physical burden or stress are associated with the rapid turnover in caregiving staff [9–11]. Due to an aging society with a shortage of caregivers, it is an urgent issue to reduce the risk of back pain and to create a more comfortable working environment for caregivers.

In recent years, robotic technology designed to reduce the physical burden on caregivers when assisting with transfers has been developed [12]. For example, the long-term use of a transfer support robot, "Hug-T1" (Fuji Corporation, Chiryu, Japan), reportedly not only reduced the physical burden on caregivers during transfer assistance but also led to an increase in verbal communication between caregivers and care recipients in a nursing facility [13,14]. Another transfer support robot, "Resyone Plus" (Panasonic AGE-FREE Co., Ltd., Kadoma, Japan), which is an electric nursing bed in which half of the bed can

be detached and functions as a fully reclining electric wheelchair, decreased the physical burden on caregivers during transfer assistance and allowed a care recipient to move easily and meet with their family members frequently in a nursing facility [15,16]. Moreover, we reported that the long-term use of wearable transfer robots, such as the "Muscle Suit" (INNOPHYS Co., Ltd., Tokyo, Japan) and "HAL®, Lumbar Type for Care Support" (CYBERDYNE, Inc., Tsukuba, Japan), led to the effective performance of transfer care tasks in nursing facilities [17]. These studies suggest that transfer support robots have the potential to decrease physical burden on caregivers and improve quality of care.

Bathing care assistance is also considered to be a physically demanding task for caregivers in relation to transfer assistance [18]. In Europe and the USA, the use of welfare equipment, such as ceiling lifts, has been implemented to support bathing activity [19,20], which has contributed to the prevention of back pain in caregivers [21,22]. However, the introduction of bathing lifts is still limited in Japan due to their financial implications and the training required for caregivers. In addition, there are technical problems that installation of these lifts into homes is limited due to their large size and the need for construction work in the bathroom [23]. Nevertheless, the act of bathing in the bathtub, as well as showering, is recognized to be important not only to keep the body clean but also as a daily relaxation, which is a popular and often habitual pastime, at least in Japan [24,25]. However, negotiating entry into a bathtub with slippery surfaces and a high threshold can be a difficult and dangerous activity for care recipients. Therefore, we believe that utilizing robotic technology to support bathing activity for care recipients and also their assistance by caregivers is highly desirable for realizing safe and secure nursing care.

In 2020, a bathing support device, "Bath Assist" (BA, the HI-LEX Corporation, Takarazuka, Japan), was developed to support the bathing activity of physically disabled or older adults (Figure 1) [26]. This is a robotic-based device that, with the simple operation of a switch, can be used to raise and lower the seat with the water pressure by connecting the shower tap to the BA controller. Since BA is a compact size (the main unit: 625 mm × 688 mm × 946 mm) and can be attached to a standard bathtub, it is expected that BA can be used not only in nursing facilities but also at home. For those who need some assistance for bathing activity, the use of BA can foster independent living with dignity. On the other hand, for caregivers, the use of BA may transform the care task from assistance to monitoring, for example, and may relieve the physical burden on caregivers.

However, it is not clear whether the use of such a bathing support device actually transforms the care task and, moreover, to what extent the physical burden on caregivers can be reduced. In order to obtain quantitative data related to the physical burden on caregivers, it would be highly significant to examine the characteristics of their muscle activities, since impaired muscle support due to excessive stress on the lumbar region, including the multifidi, erector spinae, psoas, and quadratus lumborum muscles, is suggested to be a factor in the perpetuation of lower back pain [27,28].

To clarify these issues, the present study first aimed to verify how the use of BA would transform the care task for care recipients who normally required assistance in major bathing activities, such as transfer to the shower seat, and getting into and out of the bathtub. Following this, we aimed to investigate the changes in the physical burden on caregivers with subjective questionnaires, and the changes in the characteristics of muscle activities using surface electromyography (EMG) due to the change of care. In EMG recording, we selected the erector spinae and quadratus lumborum as representative muscles supporting the lower back, and also the gluteus maximus and biceps brachii as representative muscles related to the lifting motion in hips and upper limbs of caregivers. We believe that the identification of these effects may promote the efficient and sustainable use of assistive technology such as BA.

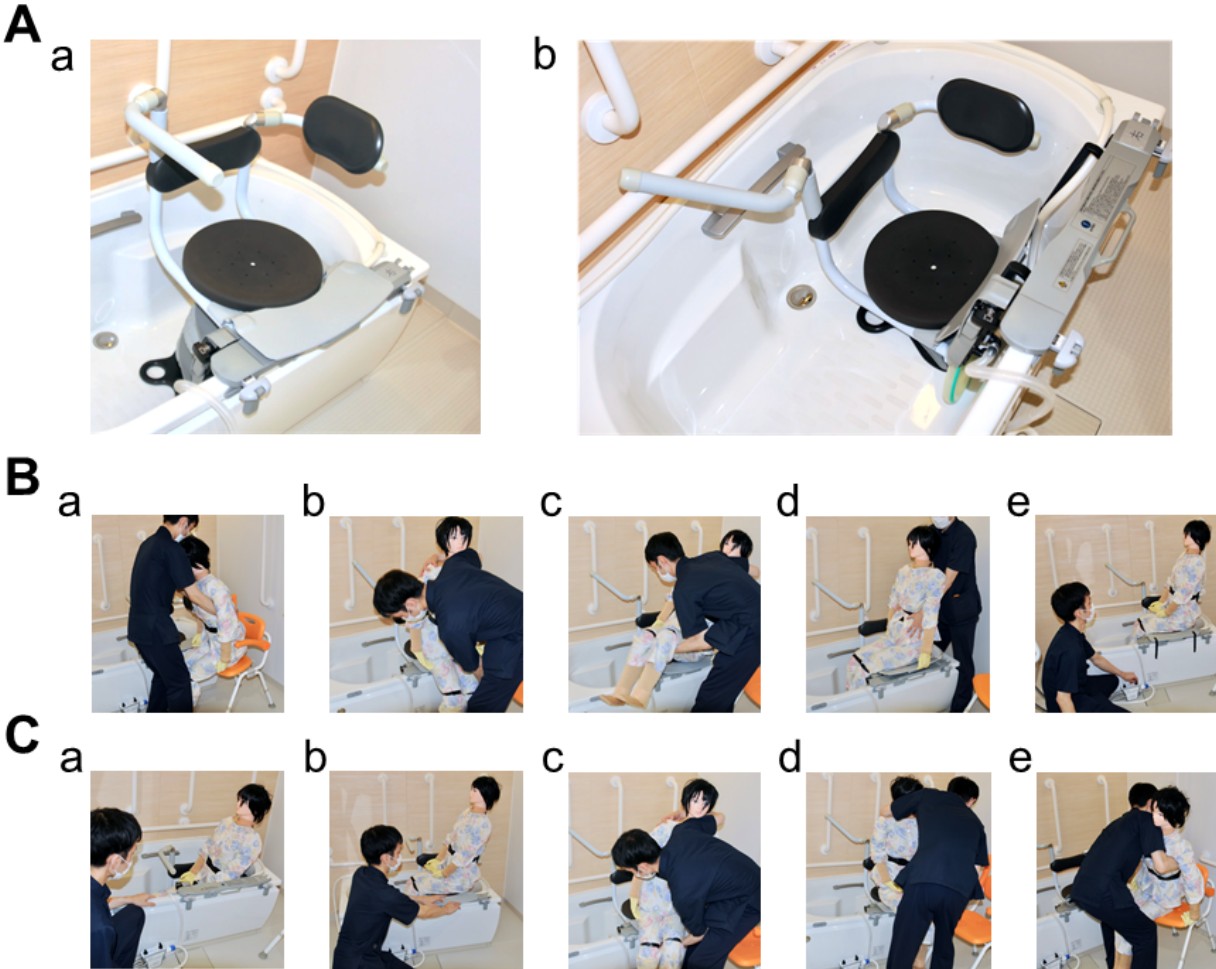

**Figure 1.** Experimental paradigm. (**A**) A picture of the bathing support device "Bath Assist" (BA). (**B**) Bathing assistance getting into the bathtub with BA. Initially, the care recipient, who is sitting in the shower chair, is lifted (a) and transferred into the seat of BA (b). Next, the seat is rotated while the care recipient's legs are moved into the bath (c). Finally, after adjusting the care recipient's posture (d), the seat is lowered using water pressure by turning the lever in the controller (e). (**C**) Bathing assistance getting out of the bathtub with BA. Initially, the seat is raised using water pressure by reversing the lever in the controller (a,b). Next, by rotating the seat of the BA, the legs of the care recipient are lifted (c) and finally transferred to the shower chair (d,e). In this figure, a life-sized mannequin was used as a patient to simulate bathing assistance.

## 2. Materials and Methods

### 2.1. Subjects

In this study, 4 physical therapists who had never used BA before were recruited as caregivers (4 males; age: $27 \pm 4$ years; work experience: $3 \pm 1$ years). In addition, 4 patients admitted to our hospital were recruited as care recipients for bathing assistance (3 males; age: $81 \pm 3$, 1 female; age: 82). Of the 4 patients, 3 required mild assistance (ID2–ID4) and 1 required moderate assistance for bathing activity (ID1). Detailed information of patients participating in this study is shown in Table 1. The caregivers and patients were paired one-to-one to simulate bathing activity without hot water in the bath, and the physical burden on the caregivers during bathing assistance was evaluated with and without the use of BA. This study was approved by the Ethics and Conflicts of Interest Committee of the National Center for Geriatrics and Gerontology, and all participants provided written informed consent.

**Table 1.** Information on patients participating in the present study.

| | | ID1 | ID2 | ID3 | ID4 |
|---|---|---|---|---|---|
| **Diagnosis** | | **Right Middle Cerebral Artery Infarction** | **Femoral Neck Fracture (Right Artificial Head Replacement)** | **Right Femoral Supraclavicular Fracture (Plate Fixation)** | **Right Parietal Lobe Infarction** |
| **Height (cm)** | | 172.8 | 166.1 | 146.1 | 152.0 |
| **Weight (kg)** | | 70.5 | 53.5 | 56.6 | 44.2 |
| FIM$^{TM}$ | Motor subscale | 60 | 78 | 70 | 76 |
| | Cognitive subscale | 29 | 31 | 32 | 33 |
| Welfare equipment used for transportation in hospital wards | | Wheelchair | Walkers | Wheelchair (For weight exemption) | Walkers |

### 2.2. BA Device

BA (weight of the main unit: approx. 8 kg; weight of the control unit: approx. 1 kg; external dimensions of the main unit: 625 mm × 688 mm × 946 mm) was developed by the HI-LEX Corporation, Japan, to support the bathing behavior of physically disabled or older adults (Figure 1A). It can be used by installing BA in the bathtub and connecting the shower hose to the BA controller. The water pressure from the shower hose can be used to raise and lower the seat of BA, mainly through lever operation. The recommended water pressure is 0.15 MPa or more. By injecting 15 L of water, the seat rises to its maximum height; it usually takes 1 to 2 min to raise or lower the seat. BA can support care recipients in the weight range of 35–80 kg. A schematic diagram of BA in operation is shown in Supplementary Materials (Figure S1). Therefore, the physical burden on caregivers in the following sequence of actions in bathing care are reduced: (1) assisting the care recipient into the bathtub, (2) lowering the care recipient into the water (as far as 80 mm from the bottom of the bath), and (3) assisting the care recipient out of the bathtub. In particular, one of the features of BA is that the seat can be rotated and the handrails in front of the seat can be held, making it easier to step into the bathtub. Note that the seat rotates freely and does not need actuation by water pressure.

### 2.3. Experimental Paradigm

The caregivers and patients were paired in a one-to-one manner in a bathroom to provide normal assistance with and without BA. When using BA, the patient sat on the seat of BA across the edge of the bathtub from a sitting position in the shower chair. Then, the patient was able to get into (Figure 1B) and out of (Figure 1C) the bathtub by asking the caregiver to operate a lever on the controller unit of BA to control the vertical movement of the seat of BA. Normal assistance without BA was performed first, and then assistance with BA was conducted once. Adequate rest periods were provided between the tasks.

In this study, surface EMG (Trigno; Delsys, Inc., Natick, MA, USA) was recorded for the caregiver during the care scenarios of getting into and out of the bathtub with and without BA (Figure 2). A total of eight wireless electrodes were attached to the left and right quadratus lumborum, erector spinae, gluteus medius, and biceps brachii muscles. The attachment locations were determined with reference to a previous study [29]. Due to the recording of EMG, the bathtub was not filled with water.

In addition, the caregivers completed a questionnaire to describe their experience of bathing assistance with and without BA by rating their perceived physical burden in a score out of 7 (i.e., 1 was the lowest and 7 was the highest physical burden). The questionnaire used in the present study is attached as Supplementary Materials.

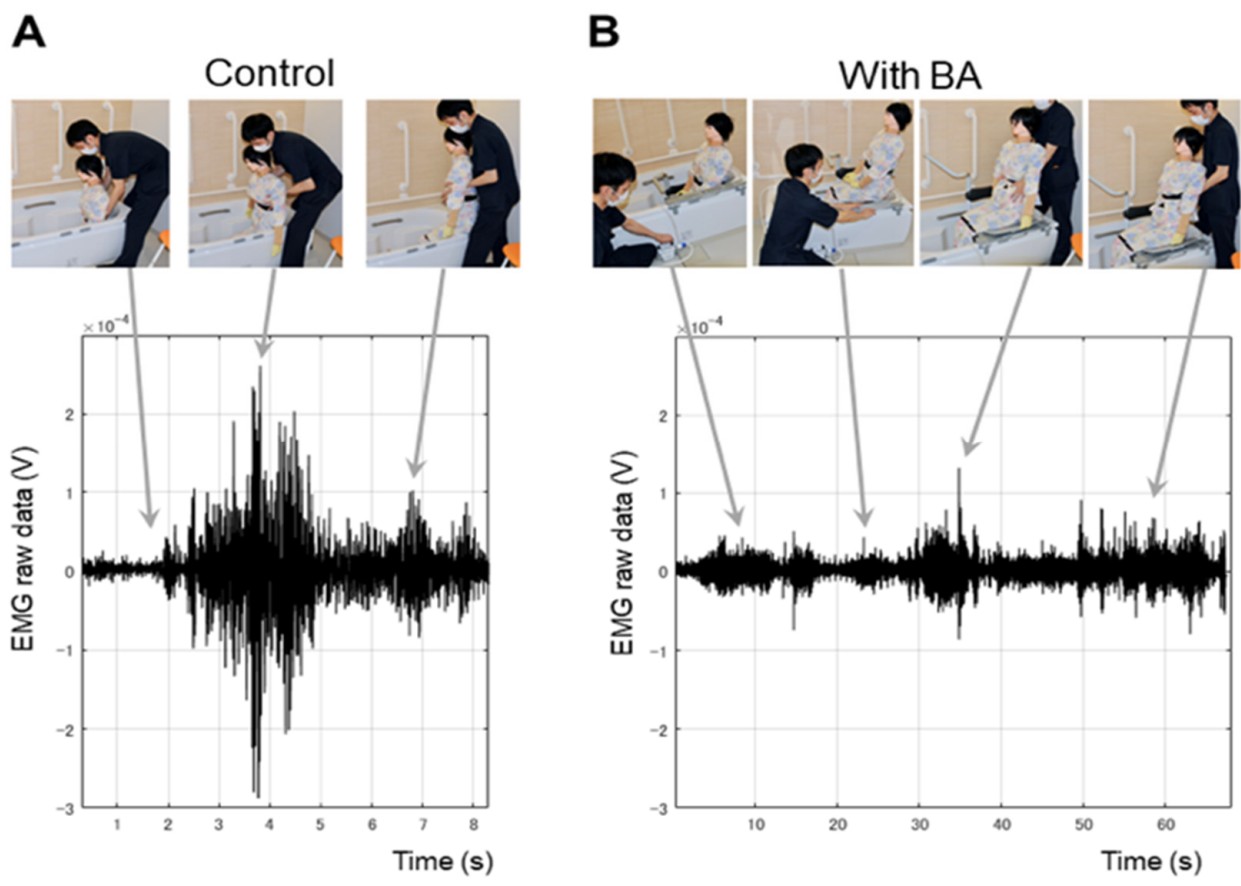

**Figure 2.** EMG recording. Raw EMG waveforms in the left quadratus lumborum muscle without BA (**A**) and with BA (**B**) when assisting a patient out of the bathtub.

### 2.4. Data Recording and Analysis

One of the authors of the present study, who carried out the experimental measurements as a physical therapist, visually classified the method of bathing care for each patient with and without BA, according to whether the patient was assisted or monitored by the caregivers in the following activities; (1) transfer to the shower seat, (2) transfer from the shower seat to the bathtub, (3) getting into the bathtub, and (4) getting out of the bathtub (Tables 2 and 3).

**Table 2.** The care requirements for bathing activities without BA.

|  | ID 1 | ID 2 | ID 3 | ID 4 |
|---|---|---|---|---|
| Transfer to shower seat | Moderate Assistance | Monitoring | Monitoring | Monitoring |
| Transfer from the shower chair to the bathtub | Moderate Assistance | Monitoring | Monitoring | Mild Assistance |
| Getting into the bathtub | Moderate Assistance | Monitoring | Monitoring | Mild Assistance |
| Getting out the bathtub | Moderate Assistance | Mild Assistance | Mild Assistance | Mild Assistance |

**Table 3.** The care requirements for bathing activities using BA.

|  | ID 1 | ID 2 | ID 3 | ID 4 |
|---|---|---|---|---|
| Transfer to shower seat | Moderate Assistance | Monitoring | Monitoring | Monitoring |
| Transfer from the shower chair to the bathtub | Moderate Assistance | Monitoring | Monitoring | Monitoring |
| Getting into the bathtub | Monitoring | Monitoring | Monitoring | Monitoring |
| Getting out the bathtub | Monitoring | Monitoring | Monitoring | Monitoring |

Since all patients required assistance getting out of the bathtub (Table 2), EMG data for caregivers from the beginning to the end of this task, with and without BA, were analyzed. The EMG sampling frequency was set to 2000 Hz. After recording, we performed off-set and full-wave rectification of the raw EMG data. The maximum values of rectified and integrated EMG over 2000 frames (1 s) were then calculated for each task and patient. Finally, rectified and integrated data were translated into percentage of maximum voluntary contraction (MVC), which was measured before the experiment based on a previously established method [30], and maximum percentage values of MVC were compared between tasks with and without BA.

## 3. Results

### 3.1. Change in Bathing Care with the Use of BA

During the experiment without using BA, we confirmed that all patients required assistance for specific movements related to bathing (Table 2). The three patients who required mild assistance were almost independent in moving from the shower chair to the seat of BA (ID2–ID4), while the patient who required moderate assistance needed partial assistance in moving from the shower chair to the seat of BA (ID1). In addition, all patients required assistance in getting out of the bathtub.

Following this, we compared the care required (moderate assistance, mild assistance, or monitoring) for typical bathing activities for each patient without BA (Table 2) and with BA (Table 3). As a result, in all patients, we confirmed the change in care from assistance to monitoring for the bathing activity of getting out of the bathtub by using BA. For two patients (ID1 and ID4) who needed assistance to get into the bathtub without BA, we confirmed the change in care from assistance to monitoring using BA. Above all, ID4 no longer needed assistance for bathing if BA was employed.

### 3.2. EMG when Assisting Patients out of the Bathub with and without the Use of BA

We then examined changes in muscle activity as measured by surface EMG, mainly in the lower back and trunk, to investigate the effect of using BA on reducing the physical burden on caregivers when assisting patients out of the bathtub. We compared the surface EMG activity of the four caregivers when assisting patients out of the bathtub with and without the use of BA. As a result, decreases in the muscle activity of left and right quadratus lumborum, erector spinae, gluteus maximus, and biceps brachii were found with the use of BA (Figure 3). For two caregivers in Figure 3A, the activity level of the quadratus limborum exceeded 100% MVC without BA (Figure 3A), indicating a high degree of exertion in the caregiver that could lead to injury.

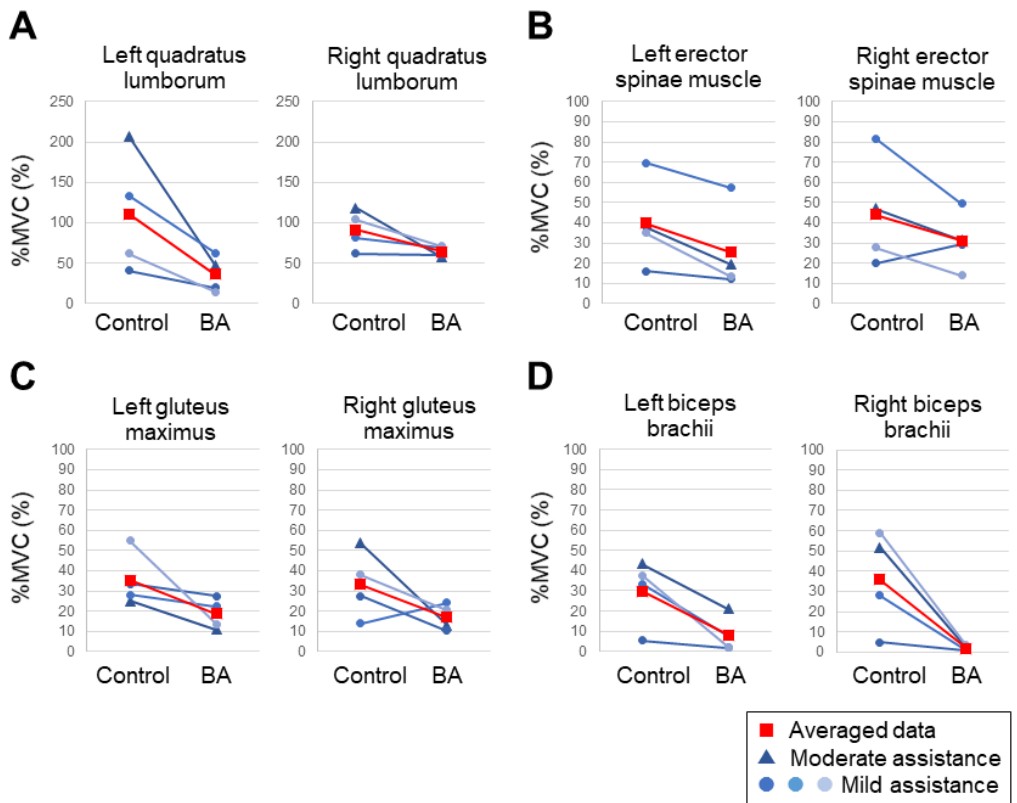

**Figure 3.** Comparison in muscle activity of left and right quadratus lumborum (**A**), erector spinae (**B**), gluteus maximus (**C**), and biceps brachii (**D**) when assisting a patient out of the bathtub without or with BA. The symbols (triangles and circles) filled in blueish colors in each graph represented EMG data obtained from the caregivers caring for patients with moderate (ID1) and mild assistance (ID2–ID4), respectively. The red square represented the averaged EMG data in each muscle.

### 3.3. Assessment of Physical Burden with and without BA Using a Questionnaire

Finally, a questionnaire was used to investigate the physical burden of each task for all caregivers with and without BA. The results showed that normal assistance without BA had a median physical burden value of 4.5, whereas the use of BA reduced the median physical burden value to 1.0 (Table 4).

**Table 4.** Results of the questionnaire on the physical burden on caregivers.

| Caregiver ID | | 1 | 2 | 3 | 4 | Median |
|---|---|---|---|---|---|---|
| Physical burden | Control | 5 | 5 | 4 | 3 | 4.5 |
| (min: 1, max: 7) | BA | 2 | 1 | 1 | 1 | 1 |

## 4. Discussion

In this study, we demonstrated that, with the use of the BA, the patients who required some assistance to get into or out of the bathtub could perform bathing activities by themselves with monitoring by the caregivers. Moreover, we found that use of BA resulted in a significant reduction in the activity of all measured muscles in the lumbar, trunk, hip, and upper limb regions in caregivers when assisting the patients out of the bathtub. More specifically, for bathing assistance without BA, the caregivers had to lift the patient while maintaining a forward leaning posture beside the bathtub. The quadratus lumborum and erector spinae muscles have roles in trunk extension and maintaining a forward leaning posture. The biceps brachii muscle is also activated when lifting heavy objects with the upper limbs, as in this experiment. However, when providing assistance with BA, the caregiver was able to raise and lower the seat by operating the lever in the BA control box.

In most cases, strenuous physical assistance was no longer necessary, and the task could be completed by simply monitoring the patient sitting on the seat of BA. Thus, there was a significant change in the method of care when using BA, from assistance to monitoring, at least when assisting the patient to get into and out of the bathtub. The results were also confirmed by the questionnaire that all of the caregivers answered, showing physical burden was decreased with BA. This accords with our results showing a substantial decrease in the activity of the recorded muscles when using BA. On the other hand, it was shown that the activity level of the quadratus limborum in one caregiver exceeded 200% MVC without BA (Figure 3A), suggesting that excessive effort was involved. Excessive stress or impairment of the lumbar muscles, such as the quadratus lumborum and multifidus muscles, is associated with an increased risk of lower back pain in younger adults [27,28]. Therefore, our findings suggest that the long-term use of BA can encourage more independent bathing activities in care recipients as well as lead to a reduction in the risk of lower back pain in caregivers, resulting in safer and more secure care. Although the results are preliminary, we have also measured the heart rate of caregivers while they are performing various types of care tasks such as transfer, toileting, bathing, and meal assistance at nursing facilities. Notably, we found that heart rate increased significantly during bathing assistance (data not shown). Since there is currently no objective, reliable indicator for estimating effort and exhaustion in caregivers, it will be necessary to accumulate evidence from various perspectives using multiple indicators related to physical burden to assess this issue.

Our results also indicate that for the patient who required a greater amount of assistance, there was a clear decrease in caregiver muscle activity during the task (Figure 3). This may suggest that caregivers assisting patients who require a higher level of assistance may benefit more from the use of BA. For example, where patients have an FIM score in the range of approximately one to four (full assistance–light assistance) for moving up and down in the bathtub, and three to five (moderate assistance–monitoring) for getting into the bathtub, a caregiver's physical burden will be effectively reduced with BA. It is also necessary to select appropriate patients for the safe use of BA. In this study, we judged that despite the small sample size, the ability to maintain a sitting position independently was required to use BA safely. However, other aspects should be assessed to ensure the safe use of BA; for example, there may be a risk of the patient losing balance backwards when assistance is required to lift a patient's legs when stepping over the edge of the bathtub.

To date, different types of bathing assist equipment, such as ceiling-type, sling-type, and floor-type lifts, have been developed to reduce the physical burden on caregivers. In Japan, the use of such equipment has been highly recommended, but only approximately 20% of nursing facilities have installed them. Their size and the need for construction work in the bathroom are likely to be the main factors preventing their introduction into homes. Other examples of welfare equipment that can be used to assist with bathing activities include bath boards, transfer benches, grab bars and stand assists (see OSHA's Guidelines for Nursing Homes [31]). However, since these devices do not have a lifting function, we consider them insufficient as benchmarks for comparison with BA. Unlike other comparable devices, BA is small enough to be installed in a normal bathtub and does not require any construction work, which may make it easier to introduce BA into homes. When installed in the home, the benefits of BA observed in this study may be helpful to family members as well as caregivers. In addition, the "robotic shower system" which automates the showering process, has been developed as a bathing aid in recent years [32]. This system would also reduce physical burden, but its installation requires construction work. As another solution, wearing an exoskeleton robot can reportedly reduce the physical burden on caregivers when providing bathing support [33,34]. Our recent study showed that in a nursing facility where HAL, a wearable transfer support robot, has been used on a daily basis for more than two years, it has been utilized effectively not only for transfer assistance but also for bathing assistance [17]. In any case, assistance in the bathtub is one of the most strenuous tasks for caregivers, involving lifting the patient and maintaining a forward leaning posture on a slippery floor [6–8]. In this sense, the future challenge is to

reduce the physical load on caregivers by incorporating these robotic technologies from various aspects.

Lastly, the present study has some limitations that merit consideration. First of all, this study was not conducted in a bathtub filled with water because of the need to perform EMG recordings. Therefore, with a bathtub containing water, there is a possibility that the physical burden on caregivers and risk of accidents may be even higher since the floor and bath surfaces are slippery, and it is difficult to provide assistance when the body of a care recipient is wet. Secondly, the number of patients in this study was limited, and further research is needed to determine the effect of the degree of required assistance on the physical burden on caregivers. Thirdly, the method used to measure MVC by surface EMG recording in this study needs further investigation. In particular, the values of MVC of the quadratus lumborum muscle exceeded 100% when the caregivers provided bathing assistance. This may be due to the fact that the method used to calculate MVC was under isometric contraction conditions. Isometric MVC has been recommended as a normalization reference value if comparisons are sought between different muscles and individuals [35]. In this review, the authors reported that EMG from MVC is reliable from some muscles, such as the elbow flexor, and is not affected by joint angle or contraction mode. However, other research has indicated large inter-individual variability in reference values for the isometric MVC method [36]. In this case, EMG from a dynamic MVC may be used [36], although it is recognized that neither method is guaranteed to reveal how active a muscle is in relation to its maximal activation capacity [37]. In this sense, the detailed electromyographic characteristics of the quadratus lumborum muscle recorded in this study are still largely unexplored and future research is needed, including the development of methods for calculating EMG reference values.

**5. Conclusions**

In conclusion, we confirmed a change in care operation from assistance to monitoring with the use of a bathing assist device, BA, which reduced both the difficulties experienced by care recipients and the physical burden on caregivers during bathing activities. In particular, muscle activities of the quadratus lumborum, erector spinae, gluteus medius, and biceps brachii reduced in caregivers. These results suggest that robotic technologies to assist with bathing care could lead to safer and more secure care, not only reducing the physical burden on caregivers but also reducing several risk factors in caregivers, such as back pain.

**Supplementary Materials:** The following supporting information can be downloaded at: https://www.mdpi.com/article/10.3390/app121910131/s1. Figure S1: Schematic diagram showing the operation of the device; Table S1: User Attitude Survey; Table S2: Psychological and Physical Burden Survey (VAS).

**Author Contributions:** Conceptualization, K.K. (Kenji Kato), K.A., K.K. (Koki Kawamura), N.I. and I.K.; methodology, K.K. (Kenji Kato), K.A., and K.K. (Koki Kawamura); validation, K.K. (Kenji Kato) and K.A.; investigation, K.A. and K.K. (Koki Kawamura); data curation, K.K. (Kenji Kato) and K.A.; writing—original draft preparation, K.K. (Kenji Kato) and T.Y.; writing—review, editing, and revising, all authors; supervision, I.K.; project administration, K.K. (Kenji Kato) and N.I.; funding acquisition, K.K. (Kenji Kato) and I.K. All authors have read and agreed to the published version of the manuscript.

**Funding:** This research was supported by AMED under Grant Nimber JP20he2002001, JSPS KAKENHI Grant Number JP19K19899, and Choju iryou kenkyu kaihatsuhi No. 19-5.

**Institutional Review Board Statement:** The study was conducted according to the guidelines of the Declaration of Helsinki and approved by the Ethics and Conflicts of Interest Committee of the National Center for Geriatrics and Gerontology (acceptance no. 1436) and all participants provided written informed consent.

**Informed Consent Statement:** Informed consent was obtained from all subjects involved in the study.

**Data Availability Statement:** The dataset analyzed in this study are not publicly available as the research team has not yet completed the analysis but are available from the corresponding author on reasonable request.

**Acknowledgments:** We thank N. Hashimoto, A. Sugiyama, M. Chiso, and H. Nakamura for their support.

**Conflicts of Interest:** The authors declare no conflict of interest.

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
