# Peer review of "Novel Bathing Assist Device Decreases the Physical Burden on Caregivers and Difficulty of Bathing Activity in Care Recipients: A Pilot Study"

_applsci, doi:10.3390/app121910131_

Round 1
Reviewer 1 Report
The paper studies labour burden reduction effectiveness for caregivers by using the Bath Assist device in bathroom. Comparison experiments are designed to show the effectiveness. EMG sensors are used to collect musule data to quantitively indicate the physical burden.
The paper is interesting. We rarely see such kind of studies. I think this study is meaningful. Comments are as follows,
1. When the recipient rotates on the BA, is it actuated by a actuator or not?
2. How large the water pressure does the device need to actuate the lift mechanism?
3. Please write the word "bathtub" consistently. Sometimes you use "bathtub". But sometimes you use "bath-tub".
Author Response
Reviewers' Comments to Author:
# Reviewer 1:
Comments and Suggestions for Authors
The paper studies labour burden reduction effectiveness for caregivers by using the Bath Assist device in bathroom. Comparison experiments are designed to show the effectiveness. EMG sensors are used to collect muscle data to quantitively indicate the physical burden. The paper is interesting. We rarely see such kind of studies. I think this study is meaningful. Comments are as follows,
Response: We thank the referees for their careful reading of the manuscript and their helpful and thoughtful suggestions. We have revised the manuscript to incorporate as many of their suggestions as we could, which has significantly improved the manuscript. We have made a number of changes that are marked in tracked changes. The editor’s and referees’ specific concerns are itemized and addressed below.
Comment 1. When the recipient rotates on the BA, is it actuated by a actuator or not?
Response: Thank you for your comment. The seat rotates freely, not actuated. We added the sentence in Method section as bellow.
Method section (line: 190)
“Note that the seat rotates freely, and does not need actuation by water pressure.”
Comment 2. How large the water pressure does the device need to actuate the lift mechanism?
Response: Thank you for your comment. Recommended water pressure is 0.15MPa or more. By injecting 15L of water, the seat rises to the maximum. When the water pressure is above 0.15MPa, it usually takes 1 to 2 minutes to raise or lower the bathtub. We added the sentence in Method section as bellow.
Method section (line: 181)
“The recommended water pressure is 0.15 MPa or more. By injecting 15 L of water, the seat rises to its maximum height; it usually takes 1 to 2 min to raise or lower the seat.”
Comment 3. Please write the word "bathtub" consistently. Sometimes you use "bathtub". But sometimes you use "bath-tub".
Response: Thank you for your comment. We have revised the word “bathtub” to be consistent throughout the manuscripts.

Reviewer 2 Report
The work explore the technological evaluation of a commercial automatic device to assist disabled patients in bathing. In this sense, the approach of the work is adequate in sections 1 and 2, which are very clear and detailed. Likewise, the description of the materials and methods used is correct. However, the scope of the study is poor and does not go beyond the logical conclusions. It does not put forward deeper hypotheses and remains at the most obvious result.
The study is relevant, but the analysis of all the data they have recorded has been poor and at least sixteen experimental subjects are required to obtain reliable statistical results. In particular, there is no objective indicator of effort or exhaustion, no comparison of the results obtained with benchmarks, and no discussion of what type of patient it is cost-effective to use the equipment with.
Author Response
# Reviewer 2:
Response: We thank the referees for their careful reading of the manuscript and their helpful and thoughtful suggestions. We have revised the manuscript to incorporate as many of their suggestions as we could, which has significantly improved the manuscript. We have made a number of changes that are marked in tracked changes. The editor’s and referees’ specific concerns are itemized and addressed below.
Comments and Suggestions for Authors
Comment 1: The work explores the technological evaluation of a commercial automatic device to assist disabled patients in bathing. In this sense, the approach of the work is adequate in sections 1 and 2, which are very clear and detailed. Likewise, the description of the materials and methods used is correct. However, the scope of the study is poor and does not go beyond the logical conclusions. It does not put forward deeper hypotheses and remains at the most obvious result.
Response: Although it is a clear result, there are also problems such as whether it is worth the busy nursing care site, such as it takes nearly two minutes to rise to the top. Nonetheless, the fact that the burden on the lower back is greatly reduced from these results does not change the fact that it is effective in reducing the burden on caregivers at nursing homes and at home, as well as expecting to reduce lower back pain.
Comment 2: The study is relevant, but the analysis of all the data they have recorded has been poor and at least sixteen experimental subjects are required to obtain reliable statistical results. In particular, there is no objective indicator of effort or exhaustion, no comparison of the results obtained with benchmarks, and no discussion of what type of patient it is cost-effective to use the equipment with.
Response:
First, as you pointed out, there does not seem to be a sufficiently reliable indicator for estimating effort or exhaustion. In the present study, we used surface electromyography as an indicator to evaluate specific muscle activity levels with the aim of estimating the physical burden on the lower back and trunk, which is the main problem of bathing assistance in nursing care. In addition, we also focused on heart rate measurement as an indicator for estimating whole-body physical burden, and are now measuring the heart rate of caregivers while they are performing various types of care tasks such as transfer, toileting, bathing, and meal assistance at nursing facilities. Although the results are preliminary, we investigated the average heart rate for each task and found that the heart rate increased significantly during bathing assistance. In any case, since there is currently no objective and reliable indicator for estimating effort and exhaustion, it will be necessary to accumulate evidence from various perspectives using many indicators related to physical burden, not limited to the present study. Thus, we added the sentence in Discussion section as bellow.
Discussion section (line: 394)
“Although the results are preliminary, we have also measured the heart rate of caregivers while they are performing various types of care tasks such as transfer, toileting, bathing, and meal assistance at nursing facilities. Notably, we found that heart rate increased significantly during bathing assistance (data not shown). Since there is currently no ob-jective, reliable indicator for estimating effort and exhaustion in caregivers, it will be necessary to accumulate evidence from various perspectives using multiple indicators related to physical burden to assess this issue.”
Second, one of the limitations of the present study is that we could not find any other suitable benchmark for a bathing assist device with lifting function that could be used at home, so the present study used a protocol that compares the results without the use of the assistive device. So far, there have been some bathing assistive devices with lifting function such as sling-type, ceiling-type, and floor-type lifts that have the ability to raise and lower the bathtub. However, it has been pointed out that these devices are sometimes difficult to install in the home because of their large size in relation to the small bathroom space and the need for construction. Other examples of welfare equipment for assisting bathtub activities include bath boards, transfer benches, grab bars, and stand assists (see OSHA's Guidelines for Nursing Homes), However, since these devices do not have the lifting function, we considered insufficient as benchmarks. Therefore, we believe that the present study has a certain value in that we were able to verify the effect of the BA, which can be installed in ordinary houses, from the viewpoint of reducing the caregiver's physical burden. In the future, it will be necessary to conduct a more detailed analysis of cost-effectiveness and patient application, comparing it with other devices. Thus, we added the sentence in Discussion section as bellow.
Discussion section (line: 414)
“To date, different types of bathing assist equipment, such as ceiling-type, sling-type, and floor-type lifts, have been developed to reduce the physical burden on caregivers.”
And also in Discussion section (line: 419)
“Other examples of welfare equipment that assist bathing activities include bath boards, transfer benches, grab bars and stand assists (see OSHA's Guidelines for Nursing Homes [30]). However, since these devices do not have a lifting function, we consider them insufficient as benchmarks for comparison with BA.”
Third, the present study indicated that at the least, the BA could change the care method from the assistance to monitoring for those who need assistance in getting into or out the bath-tub. This result implies that BA is effective in reducing caregivers’ physical burden and improving the independence of care recipients, for example, for those with relatively severe care needs and for patients who restricts some of the bathtub activities due to stroke or femoral neck fracture. Therefore, for example, the patients with FIM in the range of approximately 1-4 (full assistance - light assistance) for ascending and descending the bathtub and 3-5 (moderate assistance - monitoring) for crossing the bathtub may effectively reduce caregiver’s physical burden with the use of BA. However, since the number of subjects was limited in this study, further validation should be needed. In addition, BA costs about one thousand USD, which would low-priced compared to other devices such as sling-type, ceiling-type, and floor-type lifts that have the ability to raise and lower the bathtub. Therefore, we believe that a certain level of cost-effectiveness can be achieved by selecting and utilizing an appropriate care recipient. Thus, we added the sentence in Discussion section as bellow.
Discussion section (line: 405)
“For example, where patients have a FIM score in the range approximately 1-4 (full as-sistance - light assistance) for moving up and down in the bathtub and 3-5 (moderate assistance - monitoring) for getting into the bathtub, a caregiver’s physical burden will be effectively reduced with BA.”
Reviewer 3 Report
The paper is tacking a very important subject nowadays, namely the proper care of patients with limited motoric capabilities, namely older persons or persons with some affections which limits their capabilities in ADL. Such an activity is the hygiene where the possibility of taking a bath is critical. Further, bath can be used also for other therapeutic activities.
As shown in the paper, caregivers are subjected to a lot of stress leading to multiple accidents and muscle strains which have negative impact on both caregivers and takers.
Thus, I consider that the development of efficient solutions which take this burden away is very important, and thus I appreciate the work presented in this paper.
I would suggest some minor improvements:
(1) generally the English language is very good, I just saw in line 74 a missused verbe tense (can use should be can be used)
(2) line 162 - 163 I believe that there is a typo and the lines should be together.
Some technical aspects:
Even though there are references to previous works I'd suggest adding a figure that describes in detail the Batrhing Assistant with its components to show better how it works.
You could comment on the "relationship" between the size (weight) of the patient and the caregiver.
Regarding figure 3, could you comment on the MVC percentages, especially for the first figure where the scale for the Control group reaches over 200% which would imply a very high effort from the caregiver and could be one of the main movements/actions that lead to injuries.
Apart from that I consider this work valuable, focused on the experimental results on a system which real practical relevance and simple/quick implementation potential.
Author Response
#Reviewer3
Comments and Suggestions for Authors
The paper is tacking a very important subject nowadays, namely the proper care of patients with limited motoric capabilities, namely older persons or persons with some affections which limits their capabilities in ADL. Such an activity is the hygiene where the possibility of taking a bath is critical. Further, bath can be used also for other therapeutic activities. As shown in the paper, caregivers are subjected to a lot of stress leading to multiple accidents and muscle strains which have negative impact on both caregivers and takers. Thus, I consider that the development of efficient solutions which take this burden away is very important, and thus I appreciate the work presented in this paper.
Response: We thank the referees for their careful reading of the manuscript and their helpful and thoughtful suggestions. We have revised the manuscript to incorporate as many of their suggestions as we could, which has significantly improved the manuscript. We have made a number of changes that are marked in tracked changes. The editor’s and referees’ specific concerns are itemized and addressed below.
Comment 1. I would suggest some minor improvements: generally the English language is very good, I just saw in line 74 a missused verbe tense (can use should be can be used)
Response: Thank you for your comment. As you pointed out, we modified the verb tense in line 74.
Method section (line: 120)
“Since BA is a compact size (the main unit: 625 × 688 × 946 mm) and can be attached to a standard bathtub, it is expected that BA can be used not only in nursing facilities but also at home.”
Comment 2. line 162 - 163 I believe that there is a typo and the lines should be together.
Response: Thank you for your comment. As you pointed out, we modified the sentence in line 162 – 163.
Method section (line: 247)
“The EMG sampling frequency was set to 2,000 Hz. After recording, we performed off-set and full-wave rectification of the raw EMG data.”
Comment 3: Some technical aspects: Even though there are references to previous works I'd suggest adding a figure that describes in detail the Bathing Assistant with its components to show better how it works.
Response: As you point out, we have added a schematic figure that describes the details of the BA device and its components to show better how it works as a supplementary information.
Method section (line: 184)
“A schematic diagram of BA in operation is shown in Supplementary Figure 1.”
Comment 4: You could comment on the "relationship" between the size (weight) of the patient and the caregiver.
Response: Thank you for your comment. The lifting functions of BA can be operated mainly with a simple lever, and thus can be used by almost all caregivers. On the other hand, the size (weight) of the care recipients to whom BA can be applied is in the range of 35-80 kg. Therefore, the sentence about the relationship between the caregiver and the patient was added in Method section.
Method section (line: 183)
“BA can support care recipients in the weight range 35-80 kg.”
Comment 5: Regarding figure 3, could you comment on the MVC percentages, especially for the first figure where the scale for the Control group reaches over 200% which would imply a very high effort from the caregiver and could be one of the main movements/actions that lead to injuries.
Response: Thank you for your comment. As you pointed out, in Figure 3, the activity level of lumbar muscles (i.e. quadratus limborum) exceed 200% in condition without BA, which means that the caregiver's effort is very high and may be one of the main movements/actions that lead to injuries. Interpretations for these results were added to the Result and Discussion sections as below.
Result section (line: 328)
“For two caregivers in Figure 3A, the activity level of the quadratus limborum exceeded 100% MVC without BA (Figure 3A), indicating a high degree of exertion in the caregiver that could lead to injury.”
Discussion section (line: 387)
“On the other hand, it was shown that the activity level of the quadratus limborum in one caregiver exceeded 200% MVC without BA (Fig. 3A), suggesting that excessive effort was involved.”
Comment 6: Apart from that I consider this work valuable, focused on the experimental results on a system which real practical relevance and simple/quick implementation potential.
Response: Thank you for your careful reading and valuable comments.
Round 2
Reviewer 2 Report
The authors have not complied with the recommendations.
In particular, the design of the experiments and the results of the analysis of the experiments are very poor.
